# Targeting the Tumor Microenvironment in Acute Myeloid Leukemia: The Future of Immunotherapy and Natural Products

**DOI:** 10.3390/biomedicines10061410

**Published:** 2022-06-14

**Authors:** Christopher Hino, Bryan Pham, Daniel Park, Chieh Yang, Michael H.K. Nguyen, Simmer Kaur, Mark E. Reeves, Yi Xu, Kevin Nishino, Lu Pu, Sue Min Kwon, Jiang F. Zhong, Ke K. Zhang, Linglin Xie, Esther G. Chong, Chien-Shing Chen, Vinh Nguyen, Dan Ran Castillo, Huynh Cao

**Affiliations:** 1Department of Internal Medicine, School of Medicine, Loma Linda University, Loma Linda, CA 92354, USA; chino@llu.edu (C.H.); bpham@llu.edu (B.P.); kynishino@llu.edu (K.N.); lupu@llu.edu (L.P.); sskiwon@llu.edu (S.M.K.); 2Department of Internal Medicine, School of Medicine, University of California San Francisco–Fresno, Fresno, CA 93701, USA; daniel.park2@tu.edu; 3Department of Internal Medicine, School of Medicine, University of California Riverside, Riverside, CA 92521, USA; chiehy@medsch.ucr.edu; 4Department of Oncology/Hematology, School of Medicine, Loma Linda University, Loma Linda, CA 92354, USA; michaelnguyen@llu.edu (M.H.K.N.); simmerkaur@llu.edu (S.K.); mereeves@llu.edu (M.E.R.); dyxu@llu.edu (Y.X.); egchong@llu.edu (E.G.C.); cschen@llu.edu (C.-S.C.); 5Department of Basic Sciences, Loma Linda University, Loma Linda, CA 92354, USA; jzhonh@llu.edu; 6Department of Nutrition, Texas A&M University, College Station, TX 77030, USA; kzhang@tamu.edu (K.K.Z.); linglin.xie@tamu.edu (L.X.); 7Center for Epigenetics & Disease Prevention, Institute of Biosciences & Technology, College of Medicine, Texas A&M University, Houston, TX 77030, USA; 8Department of Biology, University of California Riverside, Riverside, CA 92521, USA; vxnguyen@gmail.com

**Keywords:** acute myeloid leukemia, tumor microenvironment, immunotherapy, natural products

## Abstract

The tumor microenvironment (TME) plays an essential role in the development, proliferation, and survival of leukemic blasts in acute myeloid leukemia (AML). Within the bone marrow and peripheral blood, various phenotypically and functionally altered cells in the TME provide critical signals to suppress the anti-tumor immune response, allowing tumor cells to evade elimination. Thus, unraveling the complex interplay between AML and its microenvironment may have important clinical implications and are essential to directing the development of novel targeted therapies. This review summarizes recent advancements in our understanding of the AML TME and its ramifications on current immunotherapeutic strategies. We further review the role of natural products in modulating the TME to enhance response to immunotherapy.

## 1. Introduction

Acute myeloid leukemia (AML) is a complex hematologic malignancy driven by the abnormal proliferation and differentiation of immature myeloid precursors [1,2]. The rapid accumulation of the AML blasts in the bone marrow (>20%) and peripheral blood consequently reduces normal hematopoiesis, resulting in life-threatening cytopenias and immunodeficiency.

Progress in deciphering the remarkable complexity and heterogeneity of AML has led to the development of selected therapies for certain AML subsets. Chemotherapy remains the mainstay treatment for AML over the decades. The use of combination of cytarabine and anthracycline, known as the “3+7 regimen”, hematopoietic stem cell transplantation (HSCT), and, more recently, the use of targeted therapies, such as hypomethylating agents (HMA, azacytidine, decitabine), venetoclax, FLT3- inhibitors, and IDH inhibitors, have resulted in promising responses in select AML subsets [3]. However, many patients develop relapsed and refractory disease, highlighting the need to better understand the AML tumor microenvironment (TME) and how it can help AML blasts evades immune surveillance.

The TME comprises a complex network of stromal cells (e.g., fibroblasts, mesenchymal and endothelial cells), immune cells (B and T lymphocytes, natural killer cells, and tumor-associated macrophages), the extracellular matrix (ECM), and secreted factors, such as cytokines [4] (Figure 1). Although the TME in AML has been recognized for many years, the critical role of the TME on disease development, progression, relapse, and resistance to therapy has only recently gained widespread attention [5]. Both preclinical experiments and clinical trials have demonstrated the potential for targeting elements of the immune microenvironment to restore proper anti-tumor immunity. In this review, we describe recent advancements in our understanding of the AML TME, current immunotherapeutic strategies under investigation, and finally potential strategies to modulate the TME using natural products to enhance response to immunotherapy. 

## 2. Composition of the AML Tumor Microenvironment

### 2.1. Mesenchymal Stromal Cells

Mesenchymal stromal cells (MSCs) are non-hematopoietic progenitors that constitute an essential component of the bone marrow (BM) niche and are well known to possess important immunomodulatory function and the ability to regulate the development and differentiation of HSCs via direct cell-to-cell contact and release of a wide array of soluble growth factors and cytokines [6,7,8]. Their ability to differentiate into other stromal components of the marrow (e.g., pericytes, myofibroblasts, BM stromal cells, osteocytes, osteoblasts, and endothelial cells) is further essential for successful allogeneic stem cell transplantation [9,10]. In the context of AML, BM- MSCs are largely considered critical contributors to tumor pathogenesis, recurrence, and resistance to chemotherapy through their ability to provide survival and anti-apoptotic signals to leukemic blasts [11,12]. Many studies have demonstrated that co-culturing AML blasts with stromal or mesenchymal stem cells result in (1) increased tumor growth [13,14], (2) aberrant phenotype expression [15,16,17], and (3) decreased sensitivity to chemotherapy [11,18,19].

### 2.2. Conventional and Regulatory T Cells

Evidence exists to suggest some degree of T lymphocyte dysfunction in AML [20]. However, their role in disease pathogenesis and the extent to which T lymphocytes influence the TME remains largely unclear. A recent study by Le Dieu et al. characterizing the T cells in the peripheral blood of newly diagnosed AML patients, surprisingly showed an increase in absolute T cell numbers [21].

Regulatory T cells (Tregs) are a CD4+ CD25+ T cell subset critical to maintain the peripheral homeostasis and central tolerance through their ability to suppress the proliferation/function of T helper cells. Over the past decade, studies have observed abnormally high levels of Tregs in the BM and peripheral blood of patients with AML compared with healthy donors [22,23]. Moreover, the presence of an elevated Treg population has been observed to correlate with worse treatment outcomes in patients with AML and the depletion of tumor-associated Tregs has been observed to improve cytotoxic T cell immunotherapy [24,25]. These findings have indicated that Tregs promote the survival of leukemia cells by inhibiting the function of effector T cells utilize direct cell-to- cell contact and the secretion of a variety of secreted factors, such as IL-35, IL10, and TGF-B [22,26].

### 2.3. Natural Killer Cells

In recent years, natural killer (NK) cells have been increasingly identified for their role in tumor immune surveillance through direct cytotoxicity, as well as indirect influence on other immune cells in the TME [27]. Using a diverse repertoire of germline-encoded inhibitory and activating receptors, NK cells can recognize cells with absent or downregulated MHC class I expression. Their importance in anti-leukemic immunity has been demonstrated by the efficacy of donor-versus-recipient NK cell alloreactivity in reducing AML relapse and reducing graft vs. host disease (GVHD) [28,29]. However, several studies have shown that NK cells are defective in AML at the time of diagnosis through the downregulation of activating receptor (e.g., natural cytotoxicity receptor family and NKG2D), upregulation of inhibitory receptors (e.g., KIR and CD94-NKGA), reduction in cytotoxic NK subtypes and impairment in NK maturation [30,31,32]. Furthermore studies have shown that AML blast ligand repertoire and correlating NK receptor expression confers better outcomes in patients undergoing chemotherapy [33,34]. The downregulation of NKG2D ligands (NKG2DL) via PARP1 in AML blasts has gained recent attention as a mechanism of NK-cell escape [35].

### 2.4. Myeloid-Derived Suppressor Cells and Tumor- Associated Macrophages

Myeloid-derived suppressor cells (MDSCs) are a heterogeneous population of CD11b+ CD33+ HLA-DR-immature myeloid cells with immunosuppressive function. The expansion of MDSCs has been well documented in AML patients, are associated with poor outcomes, tumor progression, and are believed to be involved in hampering the efficacy of immune-based therapies [36,37,38]. However, the specific mechanisms by which MDSC proliferate and contribute to the immunosuppressive TME in AML remain to be fully explored. To date, preclinical studies have suggested that MUC1 oncogene expression mediates MDSC expansion via c-myc expression [37]. Furthermore, the expression of V-domain Ig suppressor of T cell activation (VISTA), a negative regulator of tumor immune invasion has been shown to be highly expressed on MDSCs. The knockdown of VISTA was shown to reduce MDSC-mediated CD8 T cell inhibition, suggesting that VISTA may independently dampen anti-tumor response [39].

Tumor-associated macrophages (TAMs) are a major component of the TME, and the polarization to M2 phenotype in AML has been shown to subvert antitumor immunity and promote tumor progression [40,41]. Recent research has demonstrated that AML promote infiltration of TAMs in the BM and spleen in humans and mice, and that TAMs supported in vivo expansion of AML Blasts better than to macrophages from non-leukemic mice [42]. Furthermore, the expression of the transcriptional repressor Growth factor independence 1 has been shown to be crucial to macrophage polarization in the AML [42].

### 2.5. Soluble Environmental Factors

The secretion of soluble environmental factors, such as chemokines, cytokines, and growth factors, play an indispensable role in shaping the immunosuppressive milieu. Although AML blasts may prompt the secretion of pro-inflammatory cytokines, such as tumor necrosis factor-a (TNF-a), IL1b, and IL6, AML blasts also secrete immunoinhibitory factors, such as IL-10, TGF B, IL-35, and indoleamine 2,3-dioxygenase 1 (IDO1) [43,44,45]. Overexpression of these factors polarize T helper cell populations towards induced Tregs, thereby promoting T cell tolerance and leukemia immune evasion. 

Chemokines orchestrate the migration and adhesion of immune cells within the TME and are known to play a prominent role in AML progression. The expression of the chemokine CXCL12 by mesenchymal stromal cells has gained considerable attention as a potential therapeutic target in AML. CXCL12 and its receptor CXCR4 provide critical signals to prompt cell survival, adhesion, and migration. Furthermore, our lab, as well as others, have demonstrated the importance of the CXCL8-CXCR1/CXCR2 signaling pathways, which has more recently been intertwined with the upregulated expression of macrophage migration inhibitory factor (MIF) [46,47,48,49].

Several soluble metabolic factors have also been cited for their immunosuppressive capabilities in AML. Notably, the expression and release of arginase II by AML blasts has been shown to alter the immune microenvironment by enhancing arginine metabolism. By limiting arginine availability within the TME, AML blasts promote T cell exhaustion and polarize surrounding monocytes toward an suppressive M2-like phenotype [50,51]. The constitutive activation of NADPH oxidase 2 (NOX2) in >60% of AML cases has additionally been shown to result in the over production of reactive oxygen species (ROS) that promote increased glucose uptake and proliferation in AML [52,53,54].

## 3. Targeting the Tumor Microenvironment Using Immunotherapy

### 3.1. Specific Alterations in TME by AML Blasts

Alteration of the TME by AML blasts leads to increased resistance, recurrence, and progression of the disease. To better understand and discover potential treatment modalities, scientists have identified a few specific strategies deployed by AML blasts for alteration of the TME. Through the expression of immune checkpoint inhibitors, alteration of the formation of T cell immune synapses, secretion of immunoinhibitory soluble factors that affect T cell responses, and proliferation of myeloid-derived-suppressor cells (MDSC) and macrophages, AML blasts can facilitate T cell dysfunction.

NK cell dysfunction is achieved with altered expression of NK-cell ligands, such as MICA, ULBP1, ULBP2, and ULBP3 via epigenetic modifications, thus leading to alteration of the activating receptor NKG2D, which rely on those ligands, and through induction of co-inhibitory receptors in NKs, such as TIGIT, which inhibit IFN-y release. Increased TIGIT expression decreases the amount of NK cells available in marrow and is associated with poor outcomes. In addition to this, the hypoxic and inflammatory environment and metabolic reprogramming all contribute to the stromal and vascular niches in the TME, facilitating progression of the disease. Particularly, mesenchymal stromal cells (MSC) have been shown to inhibit both innate and adaptive immunity in hematological malignancies, however the theory behind how this is achieved is still being reviewed. Of note, recent literature indicates that NK cells are significantly less effective in destroying AML blasts when they are cultured with MSCs, highlighting the protective effect of the stromal microenvironment. This protective effect towards the AML blasts is hypothesized to be mediated via TLR4. Differences in MSC expression of inhibitory and pro/anti-inflammatory ligands have been noted across different clinical/cytogenetical subgroups of AML, further emphasizing the important role that they play in the pro-AML TME and the additional research needed to further elucidate their role in immunomodulation and effect on treatment with immunotherapies.

Altered immune cell homing via manipulation of the CXCL12/CXCR4 axis plays a major role in the survival, growth, and chemotherapeutic resistance of AML blasts. This key migratory axis is strongly implicated in leukemic stem cells (LSCs) relocation in bone marrow. Particularly, overexpression of CXCR4 on AML blasts portends poor prognosis, as it facilitates trafficking of malignant LSCs within BM. This effect is further amplified by decreased MSC expression of CXCL12 in AML, which reduces migration of nonmalignant stem cells in BM. Additionally, AML blasts are known to induce the downregulation and/or loss of HLA1 and HLAII leading to defective antigen presentation. Due to the complex interplay within the TME, immunotherapies have begun to be geared towards combinatorial strategies. The bone marrow microenvironment is a critical player in the NK cell response against acute myeloid leukemia in vitro [55,56].

### 3.2. Strategies to Overcome AML Resistance to Immunotherapy

Altered immune cell homing via manipulation of the CXCL12/CXCR4 axis plays a major role in the survival, growth, and chemotherapeutic resistance of AML blasts. This key migratory axis is strongly implicated in leukemic stem cells (LSCs) relocation in bone marrow. Particularly, overexpression of CXCR4 on AML blasts portends poor prognosis, as it facilitates trafficking of malignant LSCs within BM. This effect is further amplified by decreased MSC expression of CXCL12 in AML, which reduces migration of nonmalignant stem cells in BM. Additionally, AML blasts are known to induce the downregulation and/or loss of HLA1 and HLAII leading to defective antigen presentation. Due to the complex interplay within the TME, immunotherapies have begun to be geared towards combinatorial strategies [55,56].

Future directions will be focusing at overcoming multiple immunosuppressive mechanisms, as well as at targeting non-malignant components of the TME, such as stromal cells and vascular components, and enhance the immune-related effects.

### 3.3. Immunomodulation as a Hallmark of Cancer

Recognized as one of the hallmarks of cancer, immunomodulation leading to cancer cell evasion of the innate immune system has been researched in detail for the past half-century, leading to the development of novel therapies targeting the immune system and the tumor microenvironment associated with immune evasion. The immune system acts on and shapes malignant cells in a process called “immunoediting” which consists of three recognized steps: elimination, equilibrium, and escape. Elimination involves the innate and adaptive immune responses acting to eradicate developing tumors. Equilibrium, considered to be the most dangerous phase because it is the longest, is the continuous sculpting of tumor cells, which can lead to resistant variants emerging that have increasing resistance to immune pressure. Finally, in the escape phase, those resistant variants overcome the immune pressure and escape from immunomodulation, allowing them to grow uncontrollably. This final phase is where most of the current research and clinical trials are focused. Current immunotherapies try to take advantage of the three major principles of escape mechanisms: alteration of tumor or effector cells leading to the lack of antigenic recognition of malignant cells, resistance to cell death, and induction of immunotolerance and ignorance through alteration of the tumor microenvironment (TME) via secretion of immunosuppressive factors. The TME in AML and the various treatment modalities that are being trialed are the focus of this review [55,57].

### 3.4. Immune Checkpoint Blockade

The expression of inhibitory checkpoint (IC) molecules on AML blasts has been recognized as an important mechanism of immune evasion described extensively in the literature [58]. Programmed cell death ligand 1(PD-L1) is perhaps the most well-known IC molecule expressed on AML. It is encoded by the PDCD1 gene on chromosome 2q37.3 and plays a key role in maintaining self-tolerance, but also inducing CD8 T cell exhaustion in the TME by providing co-inhibitory signaling to the PD-1 receptor on T cells and promoting the expansion of Tregs [59]. The expression T cells immunoglobulin-mucin 3 (Tim-3) and C-type lectin-like inhibitory rector (CTLA)-4 have additionally been identified to be highly expressed in leukemic blasts, and experimentally shown to result in NK and T cell dysfunction (Figure 2).

Although the use of immune checkpoint inhibitors (ICIs), especially those targeting PD-1 and CTLA-4, have demonstrated remarkable efficacy in solid tumors, their application for AML remains an active area of exploration. Several studies attempted to evaluate tumor microenvironment in AML to utilize the ICI in the management of AML (Table 1). Studies have shown variable expression of inhibitory surface receptor in AML, at the time of diagnosis and relapse as compared to health controls (HC). In one study, it was noted that PD1 and OX40 are over-expressed in BM of patient with untreated and relapsed AML compared to healthy controls. There was no significant difference in PD1 expression between untreated and relapsed patients, however OX40 expression was significantly higher on all T cell subpopulations in relapsed AML compared to untreated AML. Higher expression of PD-L2 on AML blasts was also noted on untreated AML with adverse karyotype [60].

In a different study, 23 samples from patients with AML were compared with those of 30 healthy controls. PD-1 expression on CD8 + and CD4+ T cells did not differ significantly compared with healthy controls. Instead, PD-1 was upregulated in peripheral blood samples of patients with AML who relapsed after either intensive chemotherapy or allogeneic stem cell transplantation (allo-SCT) compared with those of the same patients at the time of diagnosis. Out of the two relapsed group, the upregulation of PD-1 expression on CD4 and CD8 T cells was more pronounced on its with Allo-SCT [65]. Another study demonstrated that the bone marrow tumor microenvironment in RR-AML is enriched for PD-1+ CD8+ marrow-infiltrating lymphocytes [66]. Ipilimumab showed a response in patients with hematologic malignancies that relapsed after allo-HSCT. In a multicenter phase I study, 28 patients with relapsed hematologic malignancies, including 12 patients with AML and 1 patient with MDS, were enrolled. Among five patients (23%) who achieved CR, four had AML and one had MDS [61]. 

Table 2 summaries prior clinical trials of antibody construct therapies. Ravandi et al. [67] explored the efficacy of different dosing schedules with AMG 330. A total of 55 patients with R/R AML were enrolled in 16 cohorts and administered AMG 330 on four different schedules with 0–3 dose steps prior to achieving the target dosage of 0.5–720 ug/day. Then, 100% of patients treated with AMG reported treatment-related AEs, with 90% of those AEs being attributable to AMG 330, including, but not limited to, 67% CRS, 58% dermatological issues, and 30% gastrointestinal issues. AEs that were ≥grade 3 included febrile neutropenia, CRS, skin issues, transaminitis, decreased appetite at rates of 17%, 15%, 10%, 5%, and 2%, respectively. Then, 13 patients were not able to be evaluated further for the study. Of the 42 evaluable patients, 8 responded to treatment, with 3 complete remissions (CR), 4 CR with incomplete hematological recovery (CRi), and 1 morphologic leukemia free state (MLFS).

Ranvandi et al. [70] also investigated the use of Vibecotomab, a CD123 × CD3 BiTE, at doses from 0.003 to 12.0 µg/kg, in a study that enrolled 104 patients with AML, 1 with B-cell ALL, and 1 with CML. CRS was the most common treatment related adverse event (TRAE), occurring in 60% of patients (62/104) with only 15% grade ≥3, which is reported to occur on the first dose. There was no evidence of bone marrow suppression or TLS reported. At higher doses of 0.75 µg/kg, there was an ORR of 14% (7/51, 2 Cr, 3 CRi, 2 MLFS). Stable disease was seen in 36 patients. At lower doses, no significant responses were noted, thus highlighting the higher doses needed for effective response. 

Watts et al. [71] investigated the effects of APVO436 in 46 patients with R/R AML/MDS. The most common TRAE’s seen were infusion-related reactions (28.3%) and CRS (21.7%). Transient neurotoxicity, as evidenced by headache, tremor, insomnia, memory loss and confusion were noted in 10.9% of patients. No bone marrow suppression was reported. 8/39 patients with R/R AML responded to the treatment, evidenced by stable disease, partial remissions (PR) or CR, notably 1 with >50% decrease in BM blasts, 2 with PR that deepened to CR. Overall, 3/7 evaluable patients with R/R MDS showed marrow CRs, however, this is too few to accurately gauge clinical activity. 

Subklewe et al. [68] investigated the efficacy of AMG 673 in 30 patients with R/R AML. After a median of 1.5 cycles of AMG, 90% of patients (27/30) were discontinued due to disease progression. CRS was reported in 50% of patients (15/30, 4 (13%) of which were grade ≥ 3). Other grade ≥ 3 AEs were noted to be transaminitis (17%), leukopenia (13%), thrombocytopenia (7%), and febrile neutropenia (7%). In total, 44% (12/27) patients responded to treatment as evidenced by decreased in bone marrow blasts, with 22% achieving >50% reduction. One patient achieved CRi with 85% reduction in blasts. 

Uy et al. [69] investigated the efficacy of flotetuzumab in a study of 88 patients with R/R AML. The most common TRAE was IRR/CRS, which occurred in 100% of patients, of which 60% led to dose interruptions. Other commonly seen AE’s grade ≥ 3 include anemia, thrombocytopenia, leukopenia, hypophosphatemia, and hypokalemia (28.4%, 20.5%, 18.2%, 14.8%, 13.6%, respectively). The ORR was noted to be 13.6% (12/88), with 10/88 (11.7%) with CR or CR with partial hematological recovery (CRh).

### 3.5. Bispecific T- Cell Engagers

Bispecific T-Cell Engagers (BiTEs) are bispecific antibody constructs that allow for the simultaneous binding of the CD3 receptor on T cells, and tumor-associated antigens on malignant cells. Combining the two different specificities should, in theory, activate exhausted T cells in the TME through sustained tumor antigen exposure. This highly specific process reduces off-target cytotoxicity, as the T cell will only be activated in the presence of malignant cells. Significantly, despite the downregulation of cell-surface antigens by malignant cells to evade the immune system via the downregulation of major histocompatibility complexes (MHC) and costimulatory signals, BiTEs act in an MHC-1 and signal independent manner. This is performed through the formation of a cytolytic synapse between CD8+ T cells regardless of the MHC-1 expression on tumor cells. One issue with the utilization of BiTEs Is the rapid clearance by the kidney, therefore requiring daily continuous infusion to administer the medication. Half-life-extending (HLE) BiTEs, such as AMG673, which include the fusion of an Fc domain, that leads to extension of the half-life, allowing weekly dosing of the agent [72]. The caveat to this is that additional drug retention leads to higher rates of adverse events. Figure 2 summarizes the prior clinical trials of antibody construct therapies, adapted from Allen et al. 2021 showing [1]. Further exploration and understanding of the mechanics of BiTEs have led to additional variant constructs that are still being explored (Table 3). 

Dual affinity retargeting antibodies (DARTs) have a disulfide bridge that stabilizes the structure, and results have been encouraging in AML patients, noting an objective response rate ranging between 18% and 30% [55]. Additionally, in head to head comparisons, they have yielded stronger B-cell lysis and T cell activation [73]. However, DARTs have been shown to have increased incidence of cytokine release syndrome (CRS).

Bispecific killer cell engagers (BiKEs) and trispecific killer cell engagers (TriKEs) utilize NK cells via the CD16 receptor, which when activated produces cytokines, such as IL2, that invokes a cytolytic response against the target tumor cells (Figure 3). Despite NK cells being inactivated by interaction with MHC-1, which can sometimes be expressed in AML blasts, BiKEs were able to exhibit a cytotoxic response regardless of the MHC-1 expression. TriKEs have an additional IL-15 crosslinker that allow expansion of the NK cell response and there is promising data that indicates it was able to reduce tumor burden while also sparing hematopoietic stem cells (HSCs) [73].

Some limitations of BiTE therapy include rapid renal clearance, antigen escape, and nonselective activation of the immune system. Various new constructs have been developed to overcome the rapid clearance, however they come with their own issues. Longer bioavailability leads to increased potential for adverse effects [73]. The most notable adverse events associated with these antibody constructs include cytokine release syndrome (CRS), myelosuppression, and neurotoxicity in the form of immune effector cell associated neurological syndrome (ICANS), which presents as altered mentation in the form of inattentiveness or reduced cognitive functions. ICANS is more commonly seen with CAR-T therapy but has been associated with BiTE therapy as well. Alteration of the TME via proliferation of MDSC’s and upregulation of PD-1 on activated T cells are significant barriers to using BiTEs as these both can hinder the function of T cells and reduce drug efficacy. This has led to the idea of co-administration of PDL-1 inhibitors with BiTEs, which has been effective in improving T cell proliferation and tumor cell lysis, which highlights the potential of co-administered immunotherapies to improve efficacy of these constructs. MDSCs are arising as a potential new target as they have been shown to inhibit T cell responses. Their elimination can lead to improved response rates via reduced leukemic cell burden [74].

### 3.6. Tumor Infiltrating Lymphocytes

Successful response to immunotherapy is predicated on the immunological composition of the TME. Tumor infiltrating lymphocytes (TILs) represent a heterogeneous population of effector T cells, B cells, and innate lymphoid cells recruited to the TME in response to tumor immune stimulus [75,76]. Accumulating evidence has demonstrated that increased levels of TILs correlate with a favorable microenvironment, can predict response to therapy, and improve prognosis [77,78,79]. Although tumor antigen-specific T cells should theoretically recognize and eliminate AML blasts, the presence of TILs is insufficient to inhibit tumorigenesis alone. As previously discussed, AML and its microenvironment have the capacity to both evade and suppress immune surveillance [55]. Furthermore, it has become increasingly evident that the differentiation, localization, and composition of TIL subsets is equally important for prognosis and response to treatment [80,81].

The use of engineered TILs that are expanded ex vivo and reinfused back into pre-conditioned patients has demonstrated remarkable efficacy in treating multiple solid tumors, including melanoma, colorectal, breast, and cervical cancer [82,83,84,85]. However, the application of TIL therapy in hematopoietic malignancies, including AML, remains to be fully investigated. Clinical trials for the application of marrow-infiltrating lymphocytes (MILs) in multiple myeloma have demonstrated promising efficacy in phase I/II clinical trials, highlighting the potential feasibility and efficacy for use in other hematologic tumors [86].

The clinical application of TIL therapy in AML may offer several advantages over other immunotherapies being investigated [87]. When compared to other adoptive T cell therapies, circulating TILs may be isolated from the peripheral blood rather than BM. Unlike CAR-T therapies, TIL based therapies have the advantage in that they are highly polyclonal and may respond to multiple antigens on cancer cells, rather than being limited by a defined tumor associated antigen (TAA) recognized by a CAR [88,89]. This is important to consider given that the lack of an identifiable AML-specific antigen remains a limiting factor for CAR T cell therapy in AML [90]. Allogeneic transplantation of TIL is also significantly less cytotoxic than allogeneic or CAR-T therapies given that targeted TAAs raise the risk of on-target off-tumor toxicities [91].

Using human bone marrow aspirates from AML patients, we recently demonstrated that CD3+ TILs could be isolated and expanded to clinically applicable scales using a novel ex vivo culture system [92]. Expanded TILs were immunophenotypically determined to express either CCR7^+^CD95^−^ or CD62L^+^ CD45RA^+^, which are markers for naïve T cells [93]. As proof of concept, we also showed that expanded TILs could be pharmaceutically and genetically bioengineered ex vivo to downregulate PD-1 and express lentivirus CYP27B1 gene. Finally, we showed that that TILs can cause cytotoxicity to autologous blasts ex vivo (90.6% in control vs. 1.89% in experimental groups) and are able to infiltrate the bone marrow and reside in close proximity to pre-injected autologous AML blasts of engrafted immunodeficient mice [92]. A study by Garcìa-Guerrero and colleagues alternatively demonstrated the feasibility of isolating functional tumor-reactive T cells (doublet-forming T cells) from AML patients using FACS-based cell sorting [94]. Together these results provide compelling evidence that autologous transplantation of bioengineered TILs could be used as a vehicle for gene therapy in AML.

The above studies highlight several challenges that will need to be addressed before clinical translation for use in AML. Firstly, we will need to address the reason for variability of TIL populations amongst AML patients. We observed two groups of patients that either displayed low (2.3%) or high numbers (32.6%) of CD3+ TILs, which was consistent with a recent report that CD3 TILS were preserved in 50% of patient samples [92,95]. Future experiments will also need to determine the in vivo cytotoxic and proliferative potential of TILs following transplantation in AML. It will also be necessary to demonstrate the feasibility of reversing TIL exhaustion within the context of the AML TME.

### 3.7. CAR T Cell Therapy

Chimeric antigen receptor (CAR) therapy involves engineering synthetic receptors to redirect lymphocytes, most frequently T cells, to identify and eliminate cancer cells. The extracellular domain of CARs is typically derived from the single chain variable fragment of monoclonal antibodies which bind to cancer cell antigens. Binding of CAR to the cancer cell antigen initiates intracellular domain signaling through CD3ζ which is suspected to facilitate T cell activation. Simultaneously, a costimulatory domain signaling through 4-1BB or CD28 is suspected to occur which allows T cells to have sustained anti-tumor activity [96].

The highly immunosuppressive AML microenvironment is a challenge to CAR T cell therapy. There are proposed direct and indirect mechanisms contributing to AML immune escape. AML blasts can express the enzymes arginase II and indoleamine 2,3 dioxygenase (IDO) which produce metabolites that hinder T cells while concurrently creating an environment that favors T regulatory cell and MDSC expansion which further dampen CAR T cell response and decrease proliferation. AML blasts may also express ectonucleotidases, such as CD38, CD39, and CD73, that are involved in breaking down ATP and NAD+ to adenosine; the resulting adenosine accumulation leads to T cell suppression. Notably, AML blasts have been reported to directly suppress T cell anti-tumor response by inducing reactive oxygen species (ROS) triggering T cell apoptosis [90].

Despite these challenges, success has been demonstrated in the use of CART cell therapy in AML patients. The first reported clinical trial that demonstrated CAR T cell efficacy in AML was in 2013 (Table 4). Ritchie et al. utilized a second generation CD28-ζ CAR directed against the Lewis Y antigen. The observed efficacy was limited; however, the study demonstrated CAR T durable in vivo persistence [97]. There are currently more than 20 clinical trials regarding AML and CAR T cell therapy. Targets of interest in trials are CD33, CD123, CD38, CCL1, NKG2D, Lewis Y, WT1, and CD7. CD38 is known to be expressed on most AML cells and a limited prospective study has demonstrated that CAR T-38 cells eliminated CD38 AML cells without off-target effects on lymphocytes and monocytes [98]. In an analysis of 81 primary bone marrow samples, the Lewis Y antigen has been shown to be expressed in 46% of AML cases. Moreover, CAR T therapy in a mouse model targeting Lewis Y has demonstrated prolonged survival [99]. Wilms Tumor 1 (WT1) is overexpressed in most AML patients and studies have demonstrated that increased WT1 is associated with resistance to therapy, higher incidence of relapse, and poor overall survival. WT1 CAR T cells have also exhibited the ability to lyse WT1 primary tumor cells [100]. CD33 and CD123 in particular are of interest and explained further below [101].

The most significant barrier limiting the application of CAR T cell therapy in AML is the absence of an AML-specific antigen. CD33 and CD123 are highly expressed by AML blasts and pre-clinical models have demonstrated CD33 and CD123 directed CAR T cells have highly potent anti-tumor activity [102,103]. However, as hematopoietic stem cells also express CD33 and CD123, CAR T cells are unable to differentiate between cancerous and normal cells. Several strategies have been postulated to remedy CAR T therapy limitations. One proposed mechanism is the incorporation of a “safety switch” into T cells. A traditionally utilized suicide gene in T cell therapy is HSV-tk, which allows for apoptosis of cells expressing HSV-tk upon administration of a prodrug. Use of HSV-tk is limited by immunogenicity of the viral enzyme and long latency to activation which is not effective in the setting of immediate termination of therapy in order to manage toxicity [104].

The ultimate application of CAR T therapy in AML is hypothesized in the identification of an AML specific neoantigen that can facilitate cancer cell eradication while sparing normal cells (Figure 4). Neoantigens that have been associated with AML are the metabolic enzymes IDH1 and IDH2 (estimated to be present in approximately 20% of de novo AML patients) [105,106]. Unfortunately, the proteins encoded by these mutations are expressed intracellularly and not available to CAR therapy. Dysregulated splicing may also provide a source of neoantigens. Adamia et al. reported that an estimated 33% of AML expressed genes undergo differential RNA splicing which may contribute to splice variants and potential neoantigens [107]. A subsequent study from the same group, discovered two novel splice variants, one for Flt3 and one for NOTCH 2, reported in 50% and 73% of AML patients, respectively. Of note, these splice variants were absent from healthy donors [108]. Another identified AML specific isoform is the CD44v6 variant, noted to be expressed in more than 60% of AML patients while not present in normal HPSC. CD44v6 is expressed on the cell surface therefore is available by CAR therapy and a preclinical model has shown CART cells targeting this isoform had robust anti-tumor properties [109]. As research expands into identifying CAR T cell therapy targets in AML the limitations presented by the AML TME may be addressed, thus providing patients with a novel therapeutic avenue.

## 4. Application of Natural Products in AML Therapy

### 4.1. Vitamin C

More evidence indicated that high concentrated intravenous vitamin C showed potential therapeutic effect in cell lines from cancer patients, including patients with AML. There is a dose- and time-dependent inhibition of proliferation in acute myeloid leukemia (AML) cell lines [110]. Phase I evaluation of intravenous vitamin C in combination with gemcitabine and erlotinib in cancer patients does not reveal increased toxicity [111]. As a potential cofactor of TET activity, vitamin C can induce Tet-dependent DNA demethylation, which is a type of epigenetic modification [112]. In the research of Zhao et al., low dose of vitamin C has a synergistic efficacy with decitabine in elderly patients with AML, which improved complete remission rate and prolonged overall survival [113]. Regarding RCT trial, normalization of plasma vitamin C by oral supplementation leads to an increase in the 5hmC/5mC ratio compared to placebo-treated patients [114]. Vitamin C expressed anti-proliferative effects for AML cells with both TET2 and TP53 mutations [115]. However, currently, large scale clinical trial data are required to establish the clinical anti-proliferative effect of vitamin C in AML patients.

### 4.2. Vitamin D

Through the nuclear vitamin D receptor (VDR), 1, 25(OH) 2D3 can attach to the promoter regions of target genes. Many pathways, including MAPKs, JAK/STAT, and PI3K/Akt, are related to 1, 25(OH) 2D3 [116,117,118,119,120]. MAPKs involved in 1, 25-D3-stimulated monocytic differentiation [121]. PI3K-Akt-mTOR pathway related to suppression of leukemic cell growth and involved in AML [122]. Through in vitro studies with different cell lines, multiple pathways and signals are proof related to AML cell differentiation and can be affected by VDR [123,124]. These were also proofed in some in vivo studies with mice and further confirmed the possibility of increasing survival within mice [125,126,127,128,129]. Regarding clinical experience, there was no inspiring development in this field. The previous clinical trials including AML patients and vitamin D are small scale with large variation of outcome [124,130]. In the study of Paubelle et al., combination of DFX and vitamin D significantly increased the median survival in the treatment group [131]. Low serum level of vitamin D also associated with shorter survival within AML patients [132,133]. However, the limitation of vitamin D-affect therapy is the need for high concentration of vitamin D to reach therapeutic effect, which might cause systemic side effects [134,135,136]. Several methods are possible to overcome the barriers we face currently, including possible oral supplementation [132], select patient according to receptor polymorphisms [137], gene therapy involving CYP27B1 treatment [126].

### 4.3. Vitamin B6 and Vitamin E

Compared to the board research of vitamin C and vitamin D, rare articles mentioned about vitamin B and vitamin E and those articles are in vitro studies. Tocopherols and tocotrienols are vitamin E derivatives. In Ghanem’s study, γ-tocotrienol exhibits time and dose-dependent anti-proliferative, pro-apoptotic and antioxidant effects on U937 and KG-1 cell lines (AML cell lines) [138]. Additionally, KG-1 was significantly affected at concentrations of δ-T as low as 20 μM [139]. Vitamin B6, known as pyridoxine, induces the death of primary AML cells from AML patients by activating caspase-8/3 [140]. Vitamin B6 phosphorylation regulates AML cell growth [141]. In vivo study will be needed in this new field.

### 4.4. Other Natural Products

Except for vitamins, there are lots of natural products that are proven to be related to AML and have the potential to contribute to the idea against the multi-drug resistance of AML cells. Those natural products have different anti-tumor effects, including increasing apoptosis of AML cells, inducing cell cycle arrest, anti-proliferative effects, and even reverse of drug resistance [142,143,144]. There are a few studies involved in mice in this field, including (-)-Epicatechin [145], quercetin [146], green tea [147], curcumin [148], parthenolide [149], emodin [150], compound kushen injection [151], and northern labrador tea extracts [152]. Through regulating different proteins, including Fas ligand, Bcl-2, BCL-xL, Mcl-1, histone H2AX, NF-κB, Prdxs/ROS/Trx1 signaling pathway, these natural products can induce apoptosis [145,146,147,148,149,150,151]. Not only this category, but there are also more natural products that have some in vitro evidence of effectiveness in AML cell lines (HL60, U937, KG1). Since some natural products express the possibility of toxin or side effects, and doubt about bioavailability, the method of delivery could be a possible future topic. Compared to in vitro studies and animal models, in the human population, the results of some research are disappointing. Regarding green tea, in Calgarotto’s study, despite the involvement of only 10 cases, we can only obtain the result that combined with low dose chemotherapy in the elderly are safe in this pilot study [153]. On the other hand, qinghuang powder containing arsenic, used as an alternative treatment for elderly AML patients, showed no significant difference in overall survival [154]. In conclusion, the research is limited and there is a lack of clinical effectiveness proven to happen in the human population [142,143].

## 5. Concluding Remarks

Over the past two decades, the conception of hematologic malignancies as a cell-intrinsic disorder driven by genetic and epigenetic alterations has been revised to include extrinsic mechanisms of resistance.

AML is no longer merely viewed as a genetic disease, but rather a complex interplay between leukemic cells with the surrounding cellular and humoral environment. The tumor microenvironment is an important aspect of AML biology, as it contributes to a multitude of mechanisms driving tumorigenesis and resistance to current therapies.

Immune-based therapies in AML remain of large interest, despite limited success in clinical trials. The complex immunosuppressive TME likely limits the efficacy of immune-based therapeutic strategies. Thus, it will be critical to consider the AML immune contexture so that we may develop therapies that appropriately redirect the immune microenvironment to favor the elimination of leukemic blasts. Intrinsic and extrinsic factors of AML cells, including genetic mutations and cytokine secretion, affect AML response to immunotherapies. Based on the review of the above literature, it is likely that combination therapy with ICIs, BITE, adoptive T cell therapy (CAR-T or TIL), and natural products will provide more opportunities to achieve sustained clinical remission. Early detection of resistance mechanisms and utilization of strategies to overcome resistance will be the future direction for the development of immunotherapy against AML.

## Figures and Tables

**Figure 1 biomedicines-10-01410-f001:**
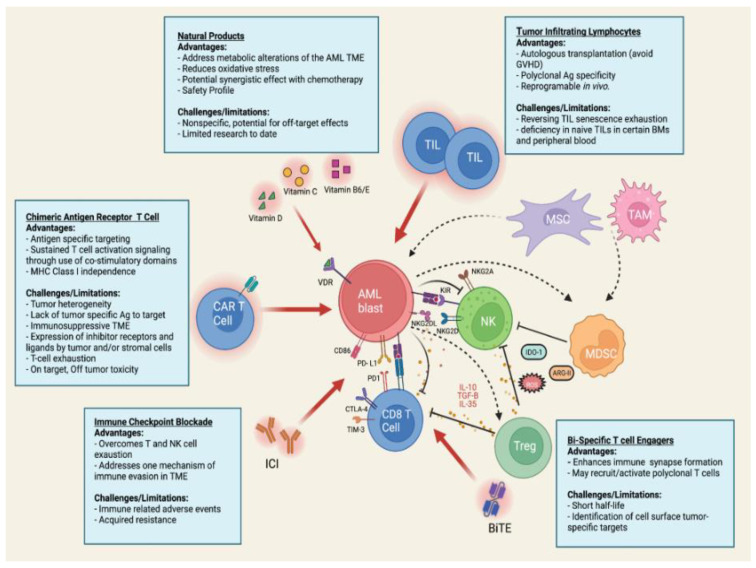
Schematic illustration summarizing the AML tumor microenvironment and current immunotherapeutic strategies. AML blasts residing in the bone marrow evade elimination through interaction with the tumor microenvironment (TME). The TME is composed of a complex network or stromal cells (fibroblasts, mesenchymal, and endothelial cells), extracellular matrix, immune cells (NK cells, TAMS, T and B lymphocytes), and the soluble factor they secrete. Together the components of the TME orchestrate the survival and proliferation of tumor cells. Approaches to target the immunosuppressive microenvironment include the use of CAR T cell, ICI, BiTE, and TIL immunotherapy or the use of natural products, such as vitamin D, C, B6, and E. Abbreviations: AML, Acute myeloid leukemia; TIL, tumor infiltrating lymphocyte; MSC, mesenchymal stromal cell; TAM, tumor associated macrophage; MDSC, myeloid derived suppressor cell; Treg, regulatory T cell, BiTE, bispecific T cell engager; ICI, immune checkpoint inhibitor; CAR, chimeric antigen receptor; NK, natural killer cell; IDO, indoleamine 2,3 dioxygenase; ARG, arginase II; ROS, reactive oxygen species; IL, interleukin; PD-L1/PD, programmed cell death/ligand 1; TIM-3, T cell immunoglobulin domain and mucin domain 3; CTLA-4, cytotoxic T-lymphocyte-associated protein 4; TGF, transforming growth factor; NKG2D, natural killer group 2 member D; KIR, Killer IG-like receptor. Image created with Biorender.com (accessed on 17 May 2022).

**Figure 2 biomedicines-10-01410-f002:**
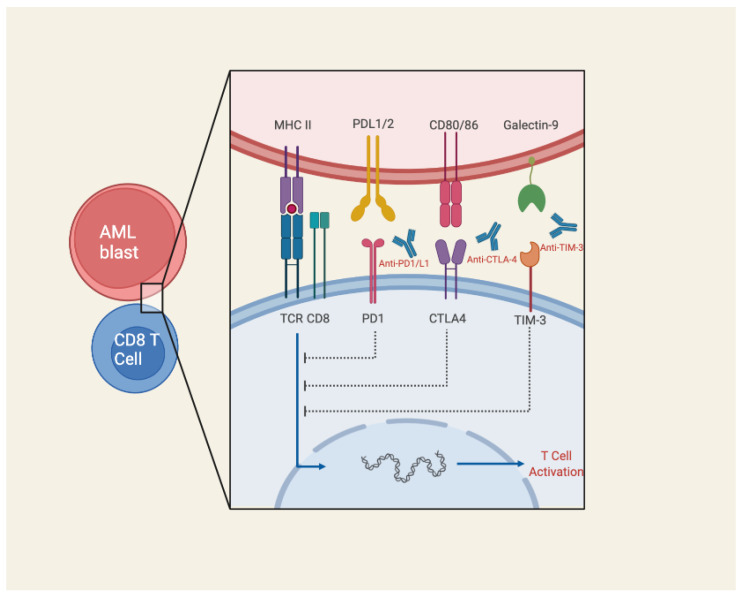
Schematic illustration of immune checkpoint inhibitor mechanism of action. Monoclonal antibodies, such as those targeting PD1/PDL-1, CTLA-4, and TIM3, block key immune checkpoint molecules typically expressed on AML blasts for the purpose of immune evasion. Image created with Biorender.com (accessed on 17 May 2022).

**Figure 3 biomedicines-10-01410-f003:**
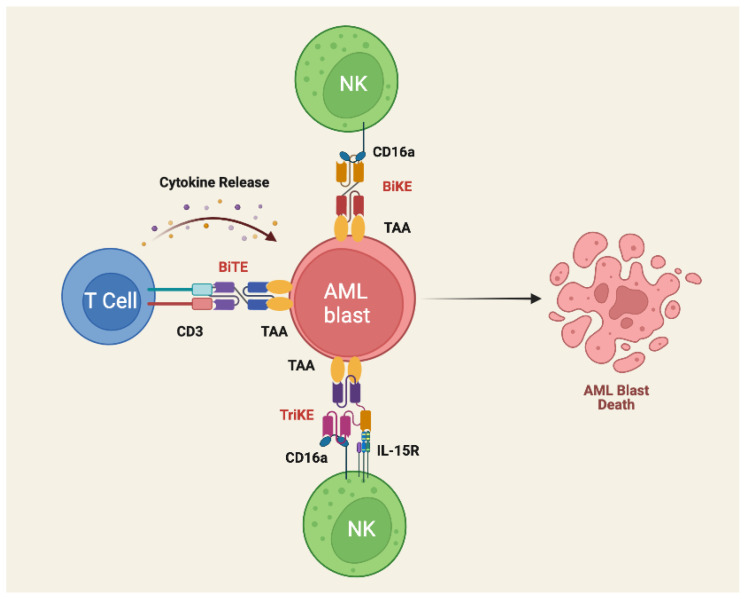
Schematic illustration of BiTE, BiKE, and TriKE structure and mechanism of action. Bispecific antibodies containing single-chain variable fragment (scFv) specific for CD3 on T cells (BiTE therapy) or CD16 on NK cells (BiKE therapy) and a specific tumor-associated antigen (TAA) are used to trigger T and NK cell activation and cytokine release. TriKE therapy similarly contains scFV specific for CD16a and TAA, but also a humanized anti-CD16 heavy chain camelid single-domain antibody (sdAb) that provides signals for NK priming, expansion, and survival. By inducing immunologic synapse formation and costimulatory signals, BiTE, BiKE, and Trike therapy can overcome immune exhaustion and improve anti-tumor activity in the setting of the AML TME. Image created with Biorender.com (accessed on 17 May 2022).

**Figure 4 biomedicines-10-01410-f004:**
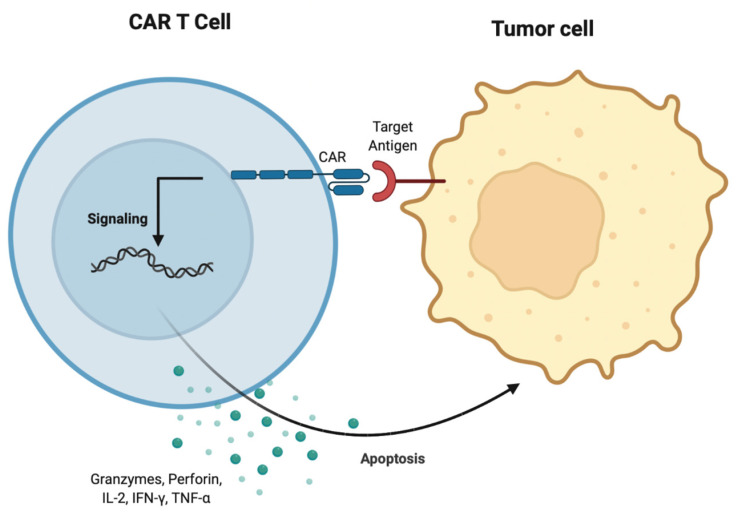
Mechanism of action of CAR T cell therapy. The T cell chimeric antigen receptor binds the target antigen on the tumor cell. This will subsequently allow T cells to mediate anti-tumor effects through the perforin and granzyme axis. Cytokines, such as IL-2, TNF-alpha, and IFN-gamma, promote T cell activation and proliferation.

**Table 1 biomedicines-10-01410-t001:** Published trials of immune checkpoint inhibitors.

Author	Phase	Intervention	Patient Population	Disease State	Outcomes
Davis et al. 2016 [61]	I/IIb	Ipilimumab	AML, NHL, HL, CML, CLL, MM, MPN, AL	Relapsed after Allo-HSCT	CR: 23% (5/28) PR: 9% (2/22)
Daver et al. 2016 [62]	I/IIb	Nivolumab + Azacitidine	AML	Relapsed after prior therapy	CR: 18% (6/51) HI: 15% (5/51)
Lindblad et al. 2018	I/II	Pembrolizumab + decitabine	AML	Relapsed after prior therapy	CR: 10% (1/10) SD:40% (4/10)
Daver et al. 2018 [63]	II	Nivolumab + Azacitidine + Ipilimumab	AML	R/R	CR/CRi: 36% (6/20) Prolonged SD:10% (2/20)
Ravandi et al. 2019 [64]	II	Idarubicin + Cytarabine + Nivolumab	AML and high risk MDS	Newly diagnosed	CR:/CRi 78% (34/44) Negative MRD: 41% (18/34)

**Table 2 biomedicines-10-01410-t002:** Prior clinical trials of antibody construct therapies.

Target	Author	Drug (Antibody Construct)	Patient Population	Outcomes
CD33	Ravandi et al. 2018/2020 [67]	AMG330 (anti-CD3 × CD33 BiTE)	55 patients with R/R-AML	Efficacy: 19% ORR (7% CR, 9% CR with incomplete hematologic recovery, 2% with morphological leukemia free state) Safety: 90% AE rate; 67% CRS (13% ≥ grade 3), nausea (20%)
CD33 (HLE *)	Subklewe et al. 2019 [68]	AMG673 (Half-Life Extended Anti-CD3 × CD33 BiTE)	30 patients with R/R-AML	Efficacy: (12/27) 44% with bone marrow blast reduction, 6 of which had >50% reduction in blasts; 1 patient with complete remission with 85% reduction Safety: 50% patients had CRS (13% ≥ grade 3), transaminitis (17%), leukopenia (13%), thrombocytopenia (7%), febrile neutropenia (7%)
CD123	Uy et al. 2021 [69]	Flotetuzumab (anti CD3 × CD123 DART)	92 R/R-AML patients	Primary induction failure or early relapse cohort (n = 30): Efficacy: 27% with CR/CRh; median OS 10.2 months among responders Safety: 100% CRS (13% ≥ grade)
CD123	Ravandi et al. 2020 [70]	Vibecotamab (XmAb14045; anti CD3 × CD123 BiTE)	104 R/R-AML, 1 B-cell ALL, and 1 CML	Efficacy: 14% ORR (4% CR); 71% SD Safety: 59% CRS (15% ≥ grade 3)
CD123	Watts et al. 2021 [71]	APVO436 (anti CD3 × CD123 BiTE)	22 R/R-AML and 6 R/R-MDS	Efficacy: 2 patients with blast reduction Safety: edema (32%), febrile neutropenia (29%), infusion reaction (21%), CRS (18%)

* Half-Life Extended (HLE).

**Table 3 biomedicines-10-01410-t003:** Summarizing the Current Trials and Targets of BiTEs, DARTs, BiKEs, and TriKEs.

Target	Drug (Antibody Construct)	Patient Population	NCT	Phase
CD33	AMV564 (CD3 × CD33 bispecific antibody)	R/R AML	NCT03144245	1
	AMG673 (CD3 × CD33 bispecific antibody)	R/R AML	NCT03224819	1
	GEM333 (CD3 × CD33 bispecific antibody)	R/R AML	NCT03516760	1
	JNJ-67571244 (CD3 × CD33 bispecific antibody)	R/R AML, MDS	NCT03915379	1
	AMG330 (CD3 × CD33 bispecific antibody)	R/R AML, Minimal Residual Disease Positive AML, MDS	NCT02520427	1
	AMV564 (CD3 × CD33 bispecific antibody)	MDS	NCT03516591	1
CD123	JNJ-63709178 (CD3 × CD123 bispecific antibody)	R/R AML	NCT02715011	1
	APVO436 (CD3 × CD123 bispecific antibody)	R/R AML, MDS	NCT03647800	1
	MGD006 (CD3 × CD123 DART)	R/R AML, MDS	NCT02152956	1 and 2
	SAR440234 (CD3 × CD123 bispecific antibody)	R/R AML, MDS, B-ALL	NCT03594955	1 and 2
	XmAb14045 (CD3 × CD123 bispecific antibody)	CD123 Expressing hematologic malignancies	NCT02730312	1
CD16/CD33	GTB-3550 (CD16/IL-15/CD33 TriKE)	R/R AML, MDS, Advanced Systemic Mastocytosis	NCT03214666	1 and 2
CD135	AMG427 (CD3 × CD135(FLT3) bispecific antibody)	R/R AML	NCT03541369	1
CLEC12A	MCLA-117 (CD3 × CLEC12A bispecific antibody)	R/R AML and newly diagnosed elderly AML	NCT03038230	1

**Table 4 biomedicines-10-01410-t004:** Ongoing clinical trials for CAR T cell therapy against AML.

Target Antigen	Population	NCT ID	Phase
CD33	R/R AML	NCT03126864	I
	R/R AML	NCT02799680	I
	R/R AML	NCT01864902	I/II
	R/R AML	NCT02944162	I/II
	R/R AML, MDS; ALL	NCT03291444	I
	R/R AML	NCT03473457	
	AML	NCT03222674	I/II
CD123	AML	NCT03585517	I
	Recurred AML after allo-HSCT	NCT03114670	I
	R/R AML	NCT03556982	I/II
	R/R AML	NCT02623582	I
	R/R AML	NCT02159495	I
	R/R AML	NCT03672851	I
	R/R AML	NCT03766126	I
	R/R AML, MDS; ALL	NCT03291444	I
	R/R AML	NCT03473457	n/a
	R/R AML	NCT03796390	I
	AML	NCT03222674	I/II
CD38	R/R AML, MDS; ALL	NCT03291444	I
	R/R AML	NCT03473457	
	AML	NCT03222674	I/II
UCART23	R/R AML	NCT03190278	I
	R/R AML, high-risk AML	NCT01864902	I
CD/123/CLL1	R/R AML	NCT03631576	II/III
CD33/CLL1	R/R AML, MDS, MPN, CML	NCT03795779	I
CCL1	AML	NCT03222674	I/II
NKG2D	AML, MDS-RAEB, MM	NCT02203825	I
	R/R AML, AML, Myeloma	NCT03018405	I/II
Lewis Y	Myeloma, AML, MDS	NCT01716364	I
WT1	R/R AML, ALL, MDS	NCT03291444	I
CD7/NK92	R/R AML	NCT03018405	I/II

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
