# Peer review of "Targeting the Tumor Microenvironment in Acute Myeloid Leukemia: The Future of Immunotherapy and Natural Products"

_biomedicines, 2022, doi:10.3390/biomedicines10061410_

Round 1
Reviewer 1 Report
The section that discussed the application of natural product is a little weak. Some grammar revision is also needed in this section.
For example: Line 484 - "Except for vitamins, there are lots of natural products that are proven to be related to AML 484 AML and have potential to contribute to the idea of against muti-drug resistance of AML 485 cells." - This needs revision for better grammar.
Some other parts of this section also needs slight revision.
Author Response
The section that discussed the application of the natural product is a little weak. Some grammar revision is also needed in this section.
For example: Line 484 - "Except for vitamins, there are lots of natural products that are proven to be related to AML 484 AML and have potential to contribute to the idea of against muti-drug resistance of AML 485 cells." - This needs revision for better grammar.
Some other parts of this section also need slight revision.
The discussion part has been revised as follows:
Except for vitamins, there are lots of natural products that are proven to be related to AML and have the potential to contribute to the idea against multi-drug resistance of AML cells. Those natural products have different anti-tumor effects, including increasing apoptosis of AML cells, inducing cell cycle arrest, anti-proliferative effects, and even reverse drug resistance 142–144. There are a few studies involved in mice in this field, including (-)-Epicatechin 145, quercetin 146, green tea 147, curcumin 148, parthenolide 149, emodin 150, compound kushen injection 151 , and northern labrador tea extracts 152. Through regulating different proteins, including Fas ligand, Bcl-2, BCL-xL, Mcl-1, histone H2AX, NF-κB, Prdxs/ROS/Trx1 signaling pathway, these natural products can induce apoptosis 145–151. Not only this category, but there are also more natural products that have some in vitro evidence of effectiveness in AML cell lines (HL60, U937, KG1). Since some natural products express the possibility of toxin or side effects and doubt about bioavailability, the method of delivery could be a possible future topic. Compared to in vitro studies and animal models, in the human population, the results of some research are disappointing. Regarding green tea, in Calgarotto’s study, despite the involvement of only 10 cases, we can only get the result that combined with low dose chemotherapy in the elderly are safe in this pilot study 153. On the other hand, arsenic-containing qinghuang powder as an alternative treatment for elderly AML patients showed no significant difference in overall survival 154. In conclusion, the research is limited and there is a lack of clinical effectiveness proven to happen in the human population 142,143.
Reviewer 2 Report
The presented review summarizes a highly relevant topic with a lot of provided information. In my opinion, it is a nice, comprehensive, and up to date review of relevant literature. The authors globally summarized the composition of tumor microenvironment in acute myeloid leukemia (AML) and the corresponding immunotherapies. Up-to-date clinical trials have been summarized and usage of natural products in AML are also discussed in this manuscript. This article is well-organized and below listed my comments:
1. This is an interesting manuscript, on some points a bit too general, owed to the huge field of immunotherapies. At some points the review could be more relevant by including detailed examples of how TME was affected in discussed immunotherapies. As the previous section introduced the composition of TME. It would be more comprehensive to involve the influence on those immune cells to modulate TME.
2. In section ‘Bispecific T-cell Engagers’, could you provide more interpretation of the outcomes regarding a few most important clinical trials?
3. To help readers easy to follow and keep focus, I strongly suggest the authors to include a few more figures to illustrate mechanism-of-action of corresponding immunotherapies.
4. As there are limitations in current developing treatments, I’d like to see more discussions in future directions to overcome problems in current strategies.
Author Response
Reviewer 2:
- This is an interesting manuscript, on some points a bit too general, owed to the huge field of immunotherapies. At some points the review could be more relevant by including detailed examples of how TME was affected in discussed immunotherapies. As the previous section introduced the composition of TME. It would be more comprehensive to involve the influence on those immune cells to modulate TME.
Altered immune cell homing via manipulation of the CXCL12/CXCR4 axis plays a major role in the survival, growth, and chemotherapeutic resistance of AML blasts. This key migratory axis is strongly implicated in leukemic stem cells (LSCs) relocation in the bone marrow. Particularly, overexpression of CXCR4 on AML blasts portends a poor prognosis, as it facilitates the trafficking of malignant LSCs within BM. This effect is further amplified by decreased MSC expression of CXCL12 in AML, which reduces the migration of nonmalignant stem cells in BM. Additionally, AML blasts are known to induce the downregulation and/or loss of HLA1 and HLAII leading to defective antigen presentation. Due to the complex interplay within the TME, immunotherapies have begun to be geared towards combinatorial strategies.
- In section ‘Bispecific T-cell Engagers’, could you provide more interpretation of the outcomes regarding a few most important clinical trials?
More interpretation of the outcomes has been discussed.
Ravandi et al 70 explored the efficacy of different dosing schedules with AMG 330. A total of 55 patients with R/R AML were enrolled in 16 cohorts and administered AMG 330 on 4 different schedules with 0-3 dose steps prior to achieving the target dosage of 0.5 – 720 ug/day. 100% of patients treated with AMG reported treatment-related AEs, with 90% of those AEs being attributable to AMG 330, including, but not limited to, 67% CRS, 58% dermatological issues, and 30% gastrointestinal issues. AEs that were ≥ grade 3 included febrile neutropenia, CRS, skin issues, transaminitis, and decreased appetite at rates of 17%, 15%, 10%, 5%, and 2% respectively. 13 patients were not able to be evaluated further for the study. Of the 42 evaluable patients, 8 responded to treatment, with 3 complete remissions (CR), 4 CR with incomplete hematological recovery (CRi), and 1 morphologic leukemia-free state (MLFS).
Ranvandi et al 73 also investigated the use of Vibecotomab, a CD123 x CD3 BiTE, at doses from 0.003 to 12.0 µg/kg, in a study that enrolled 104 patients with AML, 1 with B-cell ALL, and 1 with CML. CRS was the most common treatment-related adverse event (TRAE), occurring in 60% of patients (62/104) with only 15% grade ≥3, which is reported to occur on the first dose. There was no evidence of bone marrow suppression or TLS reported. At higher doses of 0.75 µg/kg, there was an ORR of 14% (7/51, 2 Cr, 3 CRi, 2 MLFS). Stable disease was seen in 36 patients. At lower doses, no significant responses were noted, thus highlighting the higher doses needed for effective response.
Watts et al 74 investigated the effects of APVO436 in 46 patients with R/R AML/MDS. The most common TRAE’s seen were infusion-related reactions (28.3%) and CRS (21.7%). Transient neurotoxicity, as evidenced by headache, tremor, insomnia, memory loss, and confusion was noted in 10.9% of patients. No bone marrow suppression was reported. 8/39 patients with R/R AML responded to the treatment, evidenced by stable disease, partial remissions (PR), or CR, notably 1 with >50% decrease in BM blasts, 2 with PR that deepened to CR. 3/7 evaluable patients with R/R MDS showed marrow CRs, however, this is too few to accurately gauge clinical activity.
Subklewe et al 71 investigated the efficacy of AMG 673 in 30 patients with R/R AML. After a median of 1.5 cycles of AMG, 90% of patients (27/30) were discontinued due to disease progression. CRS was reported in 50% of patients (15/30, 4 (13%) of which were grade ≥3). Other grade ≥3 AEs were noted to be transaminitis (17%), leukopenia (13%), thrombocytopenia (7%), and febrile neutropenia (7%). 44% (12/27) of patients responded to treatment as evidenced by a decrease in bone marrow blasts, with 22% achieving >50% reduction. 1 patient achieved CRi with 85% reduction of blasts.
Uy et al 72 investigated the efficacy of flotetuzumab in a study of 88 patients with R/R AML. The most common TRAE was IRR/CRS, which occurred in 100% of patients, of which 60% led to dose interruptions. Other commonly seen AE’s grade ≥3 include anemia, thrombocytopenia, leukopenia, hypophosphatemia, and hypokalemia (28.4%, 20.5%, 18.2%, 14.8%, 13.6%, respectively). The ORR was noted to be 13.6% (12/88), with 10/88 (11.7%) with CR or CR with partial hematological recovery (CRh).
- To help readers easy to follow and keep focus, I strongly suggest the authors to include a few more figures to illustrate mechanism-of-action of corresponding immunotherapies.
A few more figures have been included in the manuscript based on reviewer suggestions.
Figure 2. Schematic illustration of immune checkpoint inhibitor mechanism of action. Monoclonal antibodies, such as those targeting PD1/PDL-1, CTLA-4, and TIM3, block key immune checkpoint molecules typically expressed on AML blasts for the purpose of immune evasion.
Figure 3. Schematic illustration of BiTE, BiKE, and TriKE structure and mechanism of action. Bispecific antibodies containing single-chain variable fragment (scFv) specific for CD3 on T cells (BiTE therapy) or CD16 on NK cells (BiKE therapy) and a specific tumor-associated antigen (TAA) are used to trigger T and NK cell activation and cytokine release. TriKE therapy similarly contains scFV specific for CD16a and TAA, but also a humanized anti-CD16 heavy chain camelid single-domain antibody (sdAb) that provides signals for NK priming, expansion, and survival. By inducing immunologic synapse formation and costimulatory signals, BiTE, BiKE, and Trike therapy can overcome immune exhaustion and improve anti-tumor activity in the setting of the AML TME.
Figure 4. Mechanism of action of CAR T cell therapy. The T cell chimeric antigen receptor binds the target antigen on the tumor cell. This will subsequently allow T cells to mediate anti-tumor effects through the perforin and granzyme axis. Cytokines such as IL-2, TNF-alpha, and IFN-gamma promote T cell activation and proliferation.
- As there are limitations in current developing treatments, I’d like to see more discussions in future directions to overcome problems in current strategies.
Over the past two decades, as the fund of knowledge on hematologic malignancies continues to progress, the theory that these disease processes are a cell-intrinsic disorders driven by genetic and epigenetic alterations has been revised to include extrinsic mechanisms of resistance.
AML is no longer merely viewed as a genetic disease, but rather a complex interplay between leukemic cells with the surrounding cellular and humoral environment. The tumor microenvironment is an important aspect of AML biology, as it contributes to a multitude of mechanisms driving tumorigenesis and resistance to current therapies.
Immune-based therapies in AML remain of large interest, despite limited success in clinical trials. The complex immunosuppressive TME likely limits the efficacy of immune-based therapeutic strategies. Thus, it is critical to consider the AML immune contexture so that we may develop therapies that appropriately redirect the immune microenvironment to favor the elimination of leukemic blasts. Intrinsic and extrinsic factors of AML cells, including genetic mutations and cytokine secretion, affect AML response to immunotherapies. Based on the review of the above literature, it is likely that combination therapy with ICIs, BITE, adoptive T cell therapy (CAR-T or TIL), and natural products will provide more opportunities to achieve sustained clinical remission. Early detection of resistance mechanisms and utilization of strategies to overcome resistance will be the future direction for the development of immunotherapy against AML.
- Besides, after plagiarism check, some of the sentences share high similarity with published papers.
The highlighted paragraph has been deleted and important information has been integrated into tables.
Round 2
Reviewer 2 Report
Accept in current present